# Understanding Antimicrobial Resistance from the Perspective of Public Policy: A Multinational Knowledge, Attitude, and Perception Survey to Determine Global Awareness

**DOI:** 10.3390/antibiotics10121486

**Published:** 2021-12-04

**Authors:** SoeYu Naing, Max van Wijk, Jordi Vila, Clara Ballesté-Delpierre

**Affiliations:** 1ISGlobal, Hospital Clínic—Universitat de Barcelona, 08036 Barcelona, Spain; soeyunaing94@gmail.com (S.N.); maxxvanwijk@gmail.com (M.v.W.); JVILA@clinic.cat (J.V.); 2Centre for Biomedical Diagnosis, Department of Clinical Microbiology, Hospital Clínic, 08036 Barcelona, Spain

**Keywords:** antimicrobial resistance, knowledge, awareness, perception, governance, multinational, public health policy

## Abstract

Minimizing the effect of antimicrobial resistance (AMR) requires an adequate policy response that relies on good governance and coordination. This study aims to have a better comprehension of how AMR is understood and perceived by policy-makers and stakeholders in a multinational context. A digital survey was designed to capture the knowledge, attitudes, and perceptions (KAP) towards AMR, and it was distributed to politicians, policy advisors, and stakeholders. A total of 351 individuals from 15 different countries participated, 80% from high-income countries (HICs) and 20% from low- and middle-income countries (LMICs). The Netherlands, Spain, and Myanmar were the top 3 represented countries. Participants had sufficient knowledge regarding AMR and reported the importance of political willingness to tackle AMR. Overall, LMIC participants demonstrated better knowledge of AMR but showed poor perception and attitude towards antimicrobial use compared to HIC participants. In addition, level of education and field of expertise were significantly associated with knowledge, perception, and practices regardless of demographic characteristics. Inter-regional differences in KAP regarding AMR exist among politicians, policy advisors, and relevant stakeholders. This study captures multinational policy-maker and stakeholder mapping that can be used to propose further policy implementation on various governance levels.

## 1. Introduction

Antimicrobial resistance (AMR) is a pressing global health threat that is sometimes referred to as the “silent tsunami” or the next pandemic [1]. The misuse of antimicrobials for the prophylactic treatment of COVID-19 patients during the SARS-CoV-2 pandemic has accelerated this threat [2,3]. Moreover, the COVID-19 pandemic has highlighted that adequate policy responses, good governance, and coordination can mitigate the public health burden, suggesting that political efforts may reduce the spread and development of AMR as well. The World Health Organization (WHO) endorsed its Global Action Plan on Antimicrobial Resistance (GAP-AMR) in 2015 [4], providing countries with a framework for the development of a national action plan on AMR (NAP-AMR). As of 2020, a total of 120 countries have developed and implemented a national action plan [5], and, generally, the national plans were closely aligned with the five strategic objectives of the GAP-AMR [6]. The overall objective of the GAP-AMR is to promote a multidisciplinary approach, focusing on the inclusion of politicians, stakeholders, and scientists in tackling AMR together. In 2021, a United Nations (UN) General High-Level Interactive Dialogue on AMR called for a One Health integrative approach in AMR surveillance and implementation, research and development (R&D), budget planning, and the evaluation of AMR in the context of COVID-19 [7]. This rapid development of initiatives and growing awareness for AMR by multilateral organizations underlines that tackling AMR requires the active involvement of governments and stakeholders in a multidisciplinary context. The interest in AMR from the context of social sciences has expanded significantly over the last decade [8], but there is an urgent need for research and development (R&D) to develop new antimicrobial agents [5]. The R&D pipeline for new antimicrobials is rather slow, illustrated by the fact that there are only about 30 to 40 antimicrobials currently being tested in clinical studies [9] compared to approximately 4000 different oncology drugs that are in development [10]. This slow process is concerning as there are no treatment options left for infections with multidrug-resistant pathogens. Many pharmaceutical companies have left the antimicrobial market due to a lack of profitability, and politics-driven investments are, therefore, important to keep the antimicrobial pipeline running [11]. The lack of investment, as usually influenced by knowledge of current crises and political willingness, hinders the R&D process. Thus, it is also important to evaluate how political willingness or interest can improve the budget allocations for the R&D of new antimicrobials. 

Furthermore, call-to-action reports from multilateral organizations suggest that political willingness and interest are the required driving forces for a successful strategy in tackling the growing AMR crisis. The knowledge, attitudes, and perceptions (KAP) towards AMR from stakeholders, politicians, and policy-makers are, therefore, good indicators to assess the willingness to tackle AMR. A previous WHO report on the development of the NAP-AMR in Thailand also concluded that “strong political commitment […] will be essential for the effective implementation of the national plan” [12]. Beliefs in the cause, magnitude, and severity of the public health issue are affected by the knowledge and perception of the topic and, consequently, affect governmental responses. The poor knowledge, awareness, and perception of politicians and stakeholders toward particular public health crises can delay the progress in controlling an epidemic, as previously seen in the delayed national responses to the HIV/AIDS epidemic [13] and the ongoing COVID-19 pandemic [14]. Furthermore, lack of governmental action is commonly linked to budget constraints. However, a previous case study in Singapore has shown that the progress of AMR initiatives can be compromised regardless of resource allocation and that combating AMR is also complex in high-resource settings such as Singapore. Their findings suggest that promoting awareness among policy-makers and stakeholders is of critical importance to reduce the burden of AMR [15]. 

We have previously demonstrated a knowledge gap of infectious diseases within the general public [16], but a similar analysis from a policy context on AMR has not been done before. The awareness of AMR from the perspective of policy-makers and stakeholders has not been fully explored yet, particularly on a multinational level. Therefore, the current study aims to capture these key indicators on a multinational level, covering the KAP towards antimicrobial consumption and resistance among focal persons from government and non-government sectors. In the present study, we further investigate the variation in KAP status and AMR policy progress between high-income countries (HICs) and low-and middle-income countries (LMICs). These findings from politicians, policy-makers, and stakeholders are important in improving the political agenda toward AMR and also highlight knowledge and awareness gaps that need to be addressed. 

## 2. Results

### 2.1. Participant Characteristics

A total of 351 individuals representing 15 different countries participated in this study, with the vast majority (80.1%, n = 281) living in high-income countries (HICs) and the remaining 19.9% (n = 70) from low- and middle-income countries (LMICs). Most participants came from the Netherlands (48.7%, n = 171), followed by Spain (27.6%, n = 97) and Myanmar (9.7%, n = 34). All represented countries are shown in Figure 1.

Among the participants, 51.9% (n = 182) were female, 46.4% (n = 163) were male, and 1.7% (n = 6) did not enclose their gender. Half of the study sample (51.0%, n = 179) were between 40 and 60 years-of-age, 26.8% (n = 94) were younger than 40, and 22.2% (n = 78) were older than 60. Regarding occupational background, the vast majority (86.3%, n = 303) held positions at the government, while 12.8% (n = 45) worked for non-governmental organizations. Regarding participants working at a governmental body, 52.1% (n = 183) were working at a municipal or regional level, 18.2% (n = 64) held positions at provincial levels, and 7.1% (n = 25) were working on a national level. Educational backgrounds were mostly master/doctoral degrees (44.7%, n = 157), followed by bachelor degrees (40.7%, n = 143), and lower levels of education (14.0%, n = 49). Regarding field of expertise, 46.4% (n = 163) had a scientific background (i.e., medicine or life sciences), whereas the other half (53.6%, n = 188) had an expertise in other backgrounds. Demographic characteristics, except gender, were significantly different between HIC and LMIC subsets (*p* < 0.05). A detailed overview of all demographic characteristics is shown in Table 1. 

### 2.2. Overall Knowledge, Attitude, and Perception Scores

Cumulative scores were calculated for: (1) personal knowledge, (2) personal attitude and perception (AP), and (3) political knowledge, attitude, and perception (KAP). The mean and median scores of personal knowledge were 5.95 and 6.43 out of 10, respectively. A statistically significant difference between LMIC and HIC participants was only observed for the median score (*p* < 0.05) but not for the mean score. The median knowledge score was significantly higher for LMIC participants (7.31) compared to HIC participants (5.70). The mean and median scores of the personal AP of all participants were 6.99 and 7.50 out of 10, respectively. Both mean and median scores on personal AP were significantly different between HIC (mean of 7.31 ± 2.38, median of 7.50) and LMIC participants (mean of 5.70 ± 2.80, median of 5.83; *p* < 0.001). The political KAP scores were lower than the personal knowledge and AP scores (respectively, 2.88 vs. 5.95 and 6.99 out of 10). No statistical significance was observed between HIC and LMIC participants (see Table 2).

### 2.3. Knowledge Assessment

Only a small proportion (30.2%; 106/351) of participants knew that antimicrobial resistance is predicted to account for more deaths than cancer in the coming 30 years, and a similar proportion disagreed with the statement (26.8%; 94/351). The remaining 43.0% (151/351) reported to neither agree nor disagree or left the statement open. The level of knowledge of this AMR burden was higher in participants from LMICs (48.6%; 34/70) than those from HICs (25.6%; 72/281) (*p* < 0.001). More than half of all participants (67.2%; 236/351) correctly answered that antimicrobials cannot be used for viral infections, and a small proportion believed that antimicrobials are effective against viruses (18.6%; 65/351). Most participants (81.8%; 287/351) were well-informed that antimicrobial misuse and abuse in animal husbandry can negatively affect human health, which again differed significantly between LMIC participants (92.9%; 65/70) and HIC participants (79.0%; 222/281) (*p* < 0.05). Most participants (78.9%; 277/351) were aware that emerging resistant organisms from other countries or continents can become a problem in their own country. Fewer participants (69.8%; 245/351) knew that it is not easy to discover and produce new antimicrobials, and nearly one-third (29.9%; 105/351) neither agreed nor disagreed with this statement. The role of hygiene in tackling AMR was acknowledged by half (54.4%; 191/351) of all participants, with higher scores seen in LMIC (75.7%; 53/70) than HIC participants (49.1%; 138/281; *p* < 0.001). A detailed overview of the proportional distribution of all answer options per statement can be found in Appendix A. A bar plot of the proportion of participants with the right answers, stratified between LMICs and HICs, can be seen in Figure 2. 

A participant was considered to have a “good score” with an overall score of at least 7.0 out of 10 and a “fair score” when the overall score was at least 5.0 out of 10. The proportion with good knowledge scores declined as the education level decreased (54.8% of master/doctoral level, 41.3% of bachelor level, 22.5% in lower education graduates). Education level was significantly associated with both good and fair knowledge scores (*p* < 0.05) (see Appendix A). Upon multivariate analysis, the association between education level and good and fair scores remained significant but only between master/PhD holders and high school graduates (adjusted OR (aOR) of 0.25 for good scores (95% CI: 0.11–0.57) and 0.16 (0.07–0.37) for fair scores). Good and fair knowledge scores were positively correlated with having a scientific background (aOR 0.49 (0.31–0.79) for good scores and 0.34 (0.19–0.56) for fair scores). Furthermore, good knowledge scores were significantly associated with age (40–60 years vs. <40 years, aOR 2.21 (1.11–4.04)) and country of nationality (Spanish vs. Dutch, aOR 0.44 (0.23–0.86)) (see Table 3). 

### 2.4. Attitude and Perception Assessment

The analysis of personal attitude and perception (AP) results indicated that 38.3% (26/68) of LMIC participants consumed antimicrobials quite often (i.e., at least once every three years) compared to a significantly lower proportion of 8.8% (24/273) among HIC participants (*p* < 0.001). Regarding the completion of antimicrobial treatment, 78.8% (52/66) of LMIC participants reported that they always finish their treatment, whereas this perception and practice was significantly higher among HIC participants, with 95.2% (238/250) reporting to always finish their antimicrobial treatment (*p* < 0.001). Furthermore, only 6.5% (22/339) of all participants did not believe that antimicrobial resistance can become a health emergency issue, and this perception was more prevalent among LMIC participants (10.5%; 7/67 of LMIC participants shared this view). This difference in proportion was not statistically significant between LMIC and HIC participants. Regarding the actors that should be held responsible for tackling AMR, 30.9% (21/68) of LMIC participants reported that hospitals, veterinary clinics, and pharmaceutical industries are responsible for AMR and should solve the problem on their own. From the HIC perspective, only 10.9% (29/266) shared this view, and the difference was statistically significant (*p* < 0.001) between the two subsets. Lastly, the majority (75.2%; 254/338) reported that the current COVID-19 pandemic increased their awareness of public health and the role of governments in outbreak prevention and preparedness (see Figure 3). A detailed overview of the proportional distribution of all answer options per statement can be found in Appendix A.

Of all participants, 41.6% (146/351) and 83.8% (294/351) had a good and fair personal attitude and perception, respectively. Upon univariate and multivariate regression analysis, several factors were found to be associated with a good and fair attitude and perception (see Table 4 and Appendix A). Similarly, as described for good and fair knowledge scores, higher levels of education and scientific field of expertise remained associated with better attitude and perception after adjustment for all variables (see Table 4). Based on this multivariable regression model, LMIC participants were less likely to have a good or fair perception and attitude toward antimicrobial consumption and resistance, with an adjusted OR of 0.33 (0.14–0.75) for a good score compared to HIC participants. 

### 2.5. Political Activity and Involvement 

Only 26.7% (60/225) of HIC participants reported that a national action plan on antimicrobial resistance (NAP-AMR) had been implemented, whereas 46.9% (28/60) of LMIC participants were aware of the implementation of a NAP-AMR in their country. This difference was statistically significant (*p* < 0.01). More than half (53.1%; 34/64) of the LMIC participants reported that AMR was gaining more popularity in policies and regulations in the country, compared to a smaller proportion (39.2%; 92/143) of HIC participants. More than half of the HIC participants (56.1%; 139/248) reported that AMR interventions addressed both human and animal health, which was statistically different from the LMIC participants (40.3%) (*p* < 0.05). Accordingly, more LMIC participants (51%; 34/66) reported that AMR plans mainly focus on human health and not on the contribution of livestock. A significantly smaller proportion (33.9%; 45/218) of HIC participants agreed with this statement (*p* < 0.05) (see Figure 4).

More HIC participants (34.7%; 74/213) than LMIC participants (19.7%; 12/61) reported that hospitals in their regions had taken action to control AMR. On the contrary, almost half of the LMIC participants (47.6%; 30/63) indicated that hospitals were willing to act but lacked funding to do so. The proportion that highlighted this financial restraint was only 39.3% (83/211) among HIC participants (*p* < 0.05). When the participants were asked about the national budget and funding for AMR, less than half (35.6%; 98/275) of all participants reported that the funding and resources had increased in recent years and would be increasing in the future. Nearly everyone (80.1%; 262/327) was aware of the fact that a One Health approach should be integrated into monitoring AMR in their country, and more interventions should integrate this interdisciplinary vision. This awareness was, however, more pronounced among LMIC individuals (87.8%; 59/68) compared to HIC individuals (72.8%; 203/259) (*p* < 0.05) (See Figure 5). 

None of the sociodemographic and occupational variables were associated with overall political KAP scores (see Appendix A). The last section of the survey contained four open questions to get a better understanding of the current strategies and interventions that address AMR from within the participant’s governmental level. It aimed to identify the challenges in designing and implementing new AMR strategies. Representatives from all countries addressed the need for sufficient financial resources, promoting awareness, and educating the general public on the involved risks and the tightening of regulations and prescription behaviors to make antimicrobials less easily accessible (see Appendix A). Participants from The Netherlands especially highlighted the need for a coherent international approach rather than a national strategy to mitigate the AMR burden. Many Dutch participants also emphasized the need for more research on environmental transmission, especially the contribution of livestock and potential transmission via wastewater. Participants from Myanmar unanimously reported that the lack of knowledge and awareness of AMR among the general public, politicians, and healthcare workers should be the main focus area to address AMR, whereas there was less mention of the environmental aspects in the spread and control of AMR. One Burmese representative said that “implementing policies on AMR (e.g., legislation to reduce antimicrobials in animal feed) will be complicated; it could have an effect on the markets and economies of farmers”. Participants from Spain especially reported that surveillance programs had been implemented to better monitor the presence of antimicrobial-resistant organisms and residues in food products. In terms of interventions in place, most participants mentioned national and provincial interventions, and participants working for local authorities mainly mentioned that AMR is not being addressed on a regional level. As an example, one Spanish participant reported that “at the municipal level, we do not have direct powers on how to influence this issue”. This was in accordance with the funding source for AMR interventions, as most financial resources were from a national level in all countries. 

## 3. Discussion

The current study findings captured the personal and political knowledge, attitudes, and perceptions (KAP) of politicians and relevant stakeholders towards antimicrobial resistance (AMR) in low- and middle-income countries (LMICs) and high-income countries (HICs). The vast majority (86.3%) of the participants worked for the government, and our sample, therefore, had a good representation of civil service workers and political key players from different levels of government. Various policy evaluations and KAP assessments have previously been performed in individual countries [17], but this is the first comparative work from a multinational perspective. 

### 3.1. Personal Knowledge and Attitude towards Antimicrobial Resistance and Consumption

Current results indicated that there was a significant variation in KAP levels between participants from LMICs and HICs. While the LMIC participants displayed poor attitude and practice, they demonstrated better knowledge on AMR than the HIC participants. This difference in knowledge might be due to a sampling bias that occurred as the proportion of non-government stakeholders was higher in the LMIC (55.7%) subset compared to the HIC subset (2.1%). As an example, LMIC individuals self-reported to consume antimicrobials more regularly (38.3%) than HIC participants (8.8%). The poor attitude toward antimicrobial use among LMIC participants might be explained by the higher incidence of infections in these regions and easier access to antimicrobials since the regulations are less restricted on antimicrobial use (AMU) in LMICs compared to HICs. The Eurobarometer survey in 2018, covering over 27,000 citizens, described that almost all Europeans (93%) obtained antimicrobials via a healthcare professional [18]. Observational studies from LMICs generally show that most antimicrobials are sold without prescription [19,20]. Taken together, our study suggests that a country’s regulation of antimicrobial use is the main determinant for appropriate use, and the role of knowledge and awareness of AMR is limited when there is a lack of a regulatory framework on antimicrobial sales. However, awareness campaigns might be fruitful in LMICs when they directly target antimicrobial suppliers, as previously concluded by a study in Saudi Arabia [21]. 

Given how AMR was perceived differently in LMICs and HICs, multivariate analyses were performed to further explore explanatory factors for these differences. Notably, a scientific field of expertise and master/doctoral levels of education were the main determinants for good knowledge of AMR. These findings were similar to other studies among the general public in Sweden and Japan that showed that higher education levels yielded higher knowledge scores on antimicrobial use and AMR [22,23]. Another study from Poland furthermore showed that low levels of education were associated with poor knowledge, behavior, and attitudes toward antimicrobial use [24]. In line with these previous findings, our results highlighted that level of education was a strong predictor for personal knowledge on AMR and AMU, regardless of age, gender, or country of origin. In the case of attitude and perception, the country of origin, education, and field expertise were significant factors. Even after adjusting for other factors, including education and field of expertise, country of origin (i.e., HIC vs. LMIC) remained significantly associated with good perception and behavior. Combined, our results suggest that a background in science and having a university degree lead to the attitudes and perceptions towards antimicrobial resistance and use. However, the country’s wealth status might also play an important role, and the underlying mechanisms of how policy-makers and stakeholders perceive antimicrobial resistance remain complex. Our finding is in line with the hypothesis that a country’s regulations on antimicrobials may be one of the main determinants for the prudent use of antimicrobials on an individual level, regardless of knowledge, education, and field of expertise. 

### 3.2. Participant’s Perspectives on Political Efforts to Address AMR

The quantitative findings discussed above were reflected by the open-ended question responses that led to a better understanding of current regulations and the perspectives of politicians and stakeholders. HIC stakeholders and politicians emphasized the need for a more holistic approach integrating One Health activities. There was particular mention of addressing AMR in the field of animal husbandry, food safety, and wastewater management among HIC participants. Participants from LMICs emphasized capacity building and awareness campaigns. Interestingly, many participants working at a local governmental level mentioned that local authorities were not in the position to contribute to the AMR problem. This is a misconception, given that community-based interventions have previously been shown to reduce AMR prevalence and antimicrobial consumption [25,26]. Notably, the participants unanimously agreed that the main challenges to mitigate the AMR burden include the promotion of knowledge and awareness of the general public and stakeholders. There was a consensus on stronger regulations for antimicrobial use in all sectors among all stakeholders and civil service workers. Overall, these results provided novel insights that regional politicians demonstrated a lack of belief in regional interventions that could help mitigate the AMR burden even though political interventions had proved successful previously [27]. 

There is also a significant knowledge gap on the economic burden that AMR will cause in the future. The financial cost of AMR in the European Union alone is estimated at EUR 1.1 billion per year [28]. One representative from an LMIC shared an interesting perspective that the regulation of access to antimicrobials is not feasible since it would have a big impact on the economy of the country. The rapid emergence of AMR challenges the treatment for infections with multidrug-resistant pathogens, leaving last-resort therapeutics as the last treatment option. Antimicrobial consumption data from Vietnam has previously shown that the purchase of last-resort drugs accounted for a relatively high proportion of health budgets [29]. According to our survey responses, LMIC politicians and stakeholders might not be aware of the financial burden AMR might cause in the near future, which should, therefore, be implemented as part of awareness and knowledge campaigns. 

### 3.3. Implications for Public Health Policies

Only one out of three participants (31.1%) knew that his or her country had a national action plan on antimicrobial resistance (NAP-AMR) in place, while all represented countries had a NAP-AMR at the time of the survey [5,30]. Most participants also reported that AMR is not gaining more popularity on the political agenda and that budget allocations for AMR have not increased or are not expected to increase. On a positive note, the annual World Antimicrobial Awareness Week in November brings stakeholders, scientists, politicians, and, most importantly, the general public together to promote awareness of AMR and get updates on current interventions. These activities are well-documented and organized on a regional level in LMIC regions; for example, AMR awareness campaign materials are distributed locally and broadcasted on TV channels and social media in Myanmar [31]. AMR is a serious threat to humanity, but these numbers suggest that AMR is prioritized differently by governments around the world. The lack of national commitment to this global health problem might be due to the global and multifactorial nature of this health problem [32]. Based on our findings, the regulation of antimicrobials is the best intervention to increase the prudent use of antimicrobials. Awareness campaigns need to be aimed at politicians and influential stakeholders in order to increase budgets and political commitments. 

### 3.4. Remark on Research and Development (R&D)

The COVID-19 pandemic has highlighted that willingness and global collaboration can accelerate extraordinary discoveries in medicine. The R&D pipeline for AMR is rather slow compared to other medical fields as the discovery rate for new drugs is approximately 100-fold lower in the antibiotics industry compared to the immuno-oncology field [9,10]. More than half (64.4%) of stakeholders and policy-makers in this study reported that the funding resources allocated to reduce AMR have not increased over the years or that they do not foresee an increase in funding in the future. The R&D for new antimicrobial agents and interventions requires robust global collaboration and political willingness to increase funding and resources. Thus, understanding AMR from the perspective of public policy, as described in the current study, provides new insights into the R&D progress for AMR.

### 3.5. Methodological Considerations 

Although the data analyzed are highly relevant, it is important to consider that this study has some limitations. There was also a clear difference in demographics between the HIC and LMIC subsets, with LMIC participants mostly being stakeholders from non-government organizations, with a limited proportion of civil service workers. This sociodemographic variation might explain why the earlier observed difference between LMIC and HIC knowledge scores did not always hold during further analyses. The observation that governmental staff in LMICs were harder to reach in the current study also highlighted the challenge in obtaining information from political key players and policy-makers in these regions. Lack of access to politicians in LMICs regarding their knowledge, attitudes, and perceptions about AMR was the main limitation factor in this study. There was a selection bias for LMIC participants since they were recruited via convenience sampling. Extensive research for contact information was performed to compile mailing lists for a wide range of countries, covering more countries than included in the current study. However, many LMICs do not have an electronic governance system, which made it difficult to invite politicians from these countries. On the other hand, many HICs do not share the email addresses of politicians and policy-makers on public websites and were unwilling to provide this information via email. Consequently, there was an overrepresentation of representatives from the Netherlands (48.7%), Spain (27.6%), and Myanmar (9.7%). Initially, our study aimed to have a similar proportion of participants, and, unfortunately, the focal persons of AMR in Myanmar were no longer able to participate due to the ongoing political turmoil in Myanmar. This clear country selection bias is the major challenging factor for the generalization of the results since Dutch and Spanish participants mainly represent the HIC cohort, and half of the LMIC participants were Burmese. To overcome this selection bias, we included country of origin as a separate variable in the case of the top three represented countries (i.e., The Netherlands, Spain, and Myanmar). In general, there were small differences between the Spanish and Dutch cohorts, and the same was observed when comparing answers given by representatives from different LMIC countries. For this reason, we stratified the data based on the country’s economy (i.e., HIC vs. LMIC), although one needs to be aware that these two subsets only represent a small proportion of all HIC and LMIC countries. Furthermore, there was a self-desirability bias associated with the self-reported opinion of perceptions and attitudes toward AMU and AMR for all participants. Taking these limitations into consideration, the current study provides new and insightful data from the perspective of politicians and stakeholders worldwide. As the present work shows the differences in KAP about AMR among politicians, policy-makers, and stakeholders from LMICs and HICs, future work should perform the situational analysis of NAP-AMR to measure the progress of national action plans in these countries with AMR focal-person interviews. 

## 4. Materials and Methods

### 4.1. Design of Survey

A digital survey was developed to assess the knowledge, awareness, and perception (KAP) towards antimicrobial resistance of politicians and stakeholders on a global level (Appendix A). An informed consent page was included to inform the participants of the objective of the study and to obtain their consent. The first section of the survey collected the sociodemographic characteristics of the participants, such as gender, age, nationality, country of residence, level of education, and field of expertise. Occupational position, level of governance, political view, and political party affiliation (if applicable) were asked in order to capture the political background of every participant. Followed by demographic questions, the survey questionnaire consisted of five sections: (1) sociodemographic information, (2) statements (n = 7) that evaluated the knowledge of antimicrobial resistance, (3) statements (n = 5) that captured the attitude towards antimicrobial consumption, (4) statements (n = 14) that determined political involvement, and (5) a series of open-ended questions that capture the constraints and challenges as well as achievements and ongoing progress of political interventions aimed at combating AMR. A five-point Likert scale was provided as answer options to all close-ended statements. The survey was particularly aimed at citizens from The Netherlands, Spain, and Myanmar and was, therefore, translated into Dutch, Spanish, and Burmese. A translated survey version in French was also available for francophone countries. 

### 4.2. Participant Recruitment

Mailing lists were compiled of parliament members and governmental staff in Australia, Belgium, Canada, Curaçao, Israel, Morocco, The Netherlands, Nigeria, Spain, Singapore, and Surinam. Since Myanmar lacks an e-government system, Burmese individuals were recruited via convenience sampling, covering politicians from various governance levels and stakeholders. Ambassadors from the AMR Insight network in Mexico, Nigeria, and India (https://www.amr-insights.eu/ accessed on 27 November 2020) were considered relevant stakeholders and invited to participate in this study. A detailed overview of the number of invites sent per country can be found in Appendix A. 

### 4.3. Data Collection and Transformation

Self-administered questionnaires were distributed to the invited participants, and each response was collected using the online platform SurveyPlanet. Participants were recruited between November 2020 and March 2021. Participants were classified as working for a governmental or a non-governmental institution based on their job description. Participants that reported an expertise in medicine and/or life sciences were considered to have scientific expertise, whereas all other self-reported competencies were assigned to “other expertise”. Countries were classified as low- and middle-income (LMIC) or high-income (HIC) countries based on the 2021 World Bank classification (World Bank, 2021). 

### 4.4. Scoring System

A scoring system was adapted based on a five-point Likert scale: strongly disagree and disagree (−0.5), neither agree nor disagree (0), agree and strongly agree (+1). Prior to scoring, the answer options were reversed for negative direct statements, ensuring that correct answers were given a positive score (i.e., when strongly disagree was the right answer). Statements that were left unanswered were considered as neither agree nor disagree. The knowledge score was based on 7 statements, personal perception, and attitude scores on 6 statements; political KAP scores were based on 13 statements. The weighted cumulative scores were normalized to a maximum score of 10. Scores greater or equal to 7 were considered good scores, whereas scores greater or equal to 5 were considered fair scores. 

### 4.5. Statistical Analysis

R-studio version 1.1.447 was used for the visualization and statistical analysis of all data. Fisher’s exact test was performed to compare the proportion differences in demographic characteristics between LMICs and HICs. Univariate and multivariate logistic regressions were used to determine the relationship between the good and fair scores and participants’ demographic background.

### 4.6. Ethical Statement

Participants were asked to read the following description prior to starting the questionnaire: “This survey is intended to get insight into the awareness of politicians, decision-makers, and other related professions as well as the current state of action plans that target antibiotic resistance. [...] This survey is completely voluntary, and you can withdraw your consent at any time point. Please proceed if you have read this informed consent and agree to participate”. Participation was completely voluntary and anonymous and had no risk involved. The participant’s data were kept confidential and protected with a unique study-ID number. All participants read the description and gave informed consent to agree to participate. Only research personnel had access to data collected in this study. The current study was approved by the research board of ISGLOBAL. This study did not include medical records and, therefore, did not require an ethics committee review.

## 5. Conclusions

Overall, the current study shows how AMR is socially and politically constructed in LMICs and HICs and that politicians, policy-makers, and stakeholders face different challenges in mitigating the AMR burden. Although both LMICs and HICs showed sufficient knowledge levels of AMR in this study, the perceptions and attitudes towards antimicrobial use are associated with the country of origin (i.e., LMIC or HIC). This study identifies that awareness interventions targeting politicians and stakeholders are lacking and that more political action is required to combat the AMR crisis.

## Figures and Tables

**Figure 1 antibiotics-10-01486-f001:**
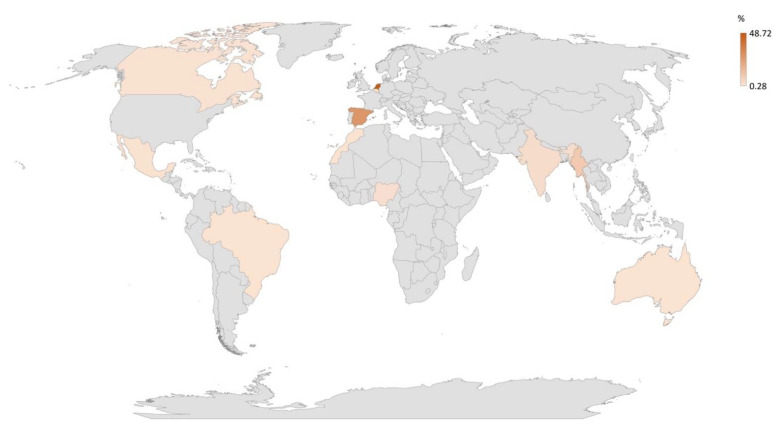
Spatial plot of all countries represented in current multinational study: the Netherlands (48.7%, n = 171), Spain (27.6%, n = 97), Myanmar (9.7%, n = 34), India (3.1%, n = 11), Nigeria (2.6%, n = 9), Mexico (1.7%, n = 6), Morocco (1.4%, n = 5), Australia (1.1%, n = 4), Brazil (1.1%, n = 4), Belgium (0.9%, n = 3), Canada (0.9%, n = 3), Curaçao (0.3%, n = 1), Guatemala (0.3%, n = 1), Panama (0.3%, n = 1), and Singapore (0.3%, n = 1).

**Figure 2 antibiotics-10-01486-f002:**
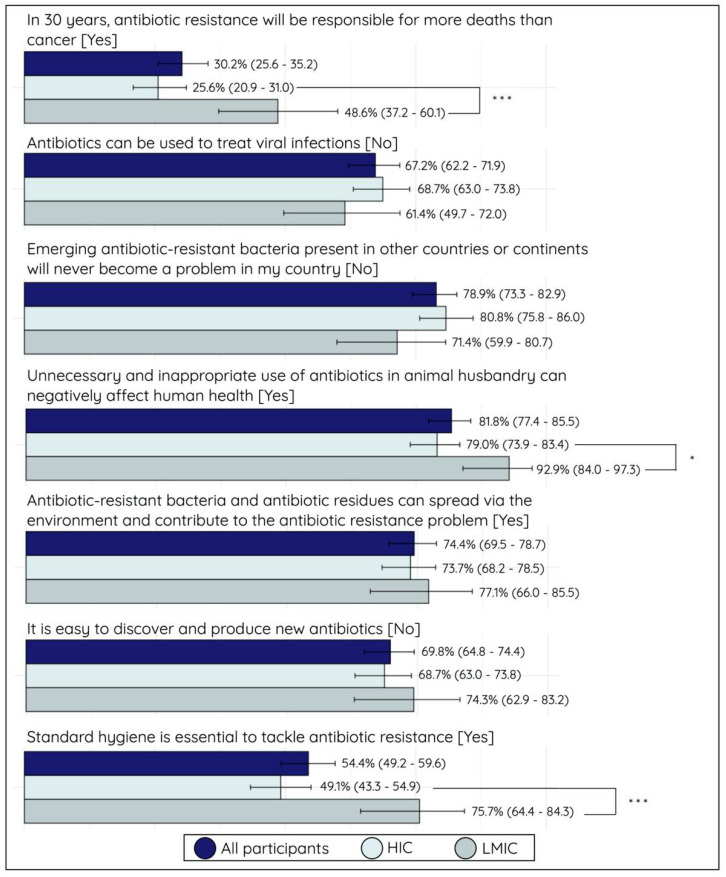
Bar plot (proportion of questions correctly answered, with 95% confidence intervals) of knowledge of questions of all participants, stratified for high-income country (HIC) and low- and middle-income country (LMIC) participants. The correct answer option (yes/no) is shown behind each statement, with “yes” corresponding to agree and strongly agree and “no” to disagree and strongly disagree with the survey statements. Significance: *** *p* < 0.001, and * *p* < 0.05.

**Figure 3 antibiotics-10-01486-f003:**
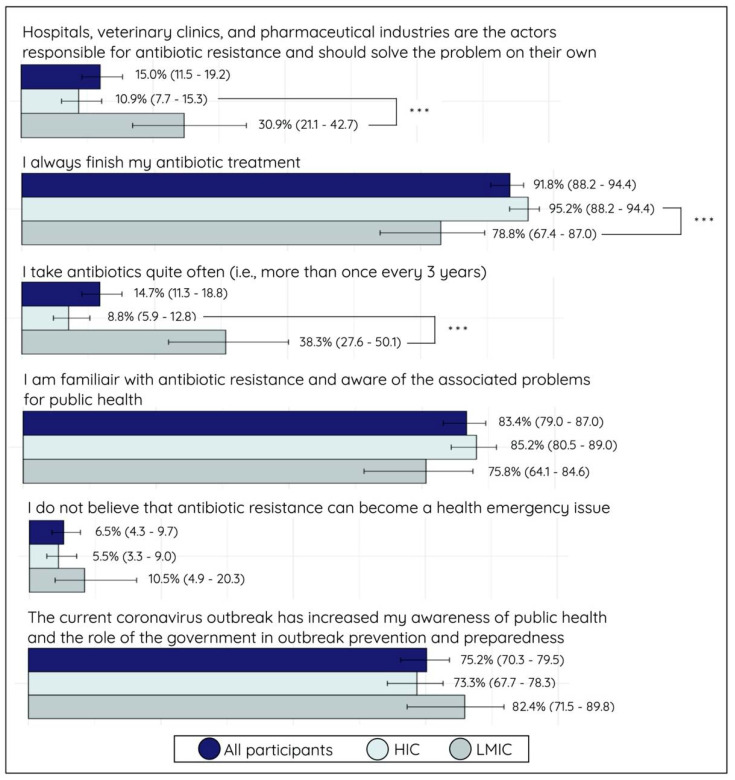
Bar plot (proportion (%) with 95% confidence interval) of personal attitude and perception statements from all participants, stratified for high-income country (HIC) and low- and middle-income country (LMIC) participants. The proportion represents participants who answered the statement with agree or strongly agree. Significance: *** *p* < 0.001.

**Figure 4 antibiotics-10-01486-f004:**
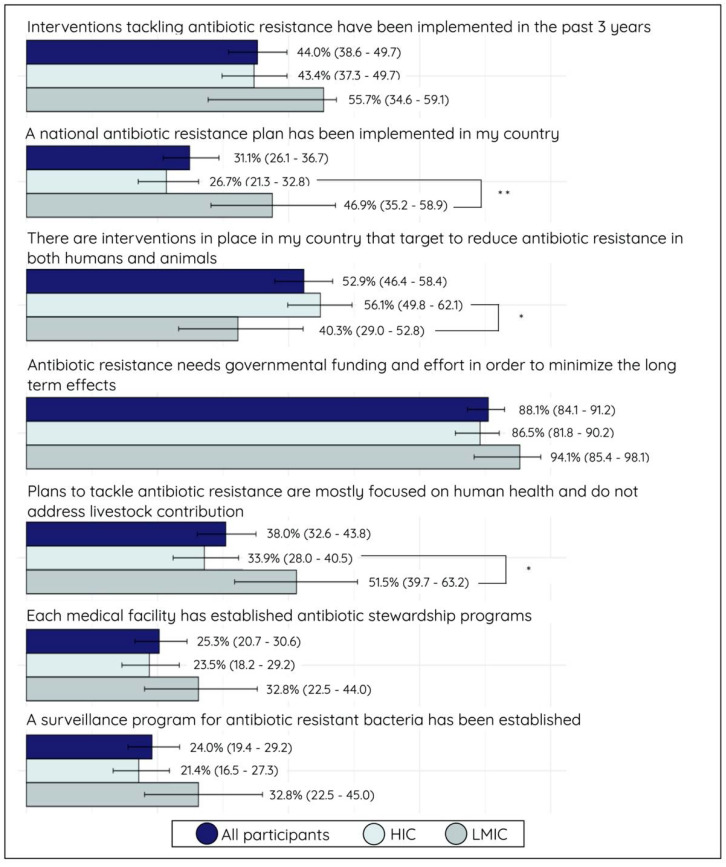
Bar plot (proportion (%) with 95% confidence interval) on the first part of statements assessing the political knowledge, attitudes, and perceptions (KAP) of all participants, stratified for high-income country (HIC) and low- and middle-income country (LMIC) participants. The proportion represents participants who answered the statement with agree or strongly agree. Significance: ** *p* < 0.01, and * *p* < 0.05.

**Figure 5 antibiotics-10-01486-f005:**
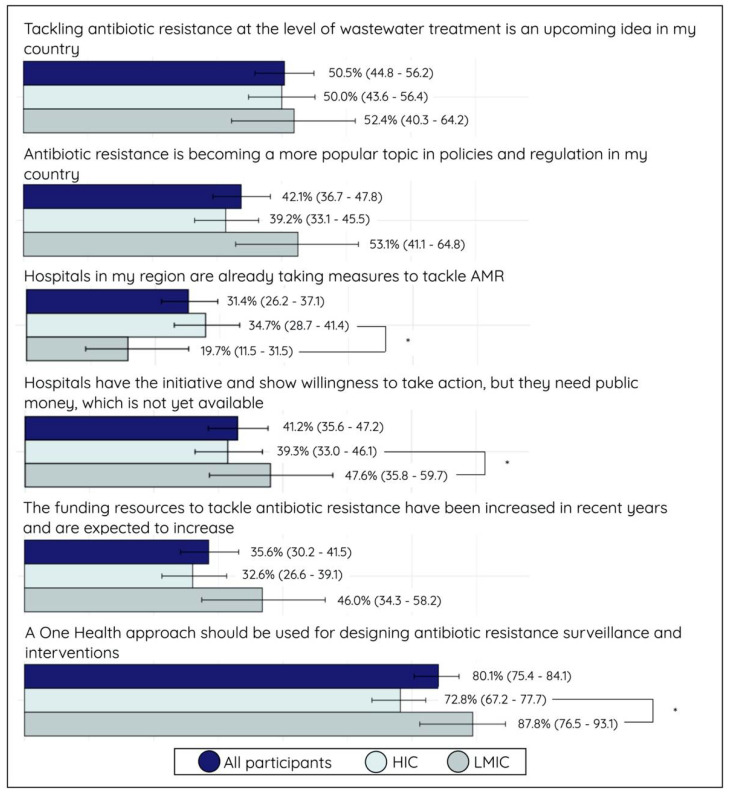
Bar plot (proportion (%) with 95% confidence interval) on the second part of statements assessing the political knowledge, attitudes, and perceptions (KAP) of all participants, stratified for high-income country (HIC) and low- and middle-income country (LMIC) participants. The proportion represents participants who answered the statement correctly with agree or strongly agree. Significance: * *p* < 0.05.

**Table 1 antibiotics-10-01486-t001:** Sociodemographic description of all participants, stratified for low- and middle-income (LMICs) and high-income countries (HICs). Differences between subsets were analyzed by Fisher’s exact test. Significance: *** *p* < 0.001, and ** *p* < 0.01.

	Complete Dataset % (N)	HICs % (N)	LMICs % (N)	Fisher Exact
Total	351	80.1% (281)	19.9% (70)	
**Top 3 nationalities ^A^**	NLD 48.7% (171)	NLD 60.9% (171)	MMR 48.6% (34)	
	ESP 27.6% (97)	ESP 34.5% (97)	IND 15.7% (11)	
	MMR 9.7% (34)	AUS 1.4% (4)	NGA 12.9% (9)	
**Gender**				
Female	51.9% (182)	40.8% (15)	45.7% (32)	0.252
Male	46.4% (163)	44.5% (125)	54.3% (38)	
Undisclosed	1.7% (6)	2.1% (6)	0.0% (0)	
**Age**				
Mean ± standard deviation	49.3 ± 13.3	52.2 ± 12.0	37.8 ± 12.3	
Median [IQR]	52.0 [21.5–62.5]	55 [46.5–63.5]	33 [23.3–42.8]	
**Age group**				
<40	26.8% (94)	16.4% (46)	68.6% (49)	<0.001 ***
40–60	51.0% (179)	57.3% (161)	25.7% (18)	
>60	22.2% (78)	26.3% (74)	5.7% (4)	
**Position duration**				
<1 year	7.7% (27)	8.2% (23)	5.7% (4)	<0.001 ***
1–3 years	30.2% (106)	33.5% (94)	17.1% (12)	
3–5 years	17.1% (60)	12.5% (35)	35.7% (25)	
5–10 years	19.7% (69)	19.2% (54)	21.4% (15)	
>10 years	25.4% (89)	26.7% (75)	20.0% (14)	
Educational background				
Master/PhD	44.7% (157)	39.5% (111)	65.7% (46)	<0.001 ***
Bachelor	40.7% (143)	42.7% (120)	32.9% (23)	
Lower levels	14.0% (49)	17.1% (48)	1.4% (1)	
Unknown	0.6% (2)	0.7% (2)	0.0% (0)	
**Expertise**				
Scientific	46.4% (163)	42.7% (120)	61.4% (43)	0.008 **
Other background	53.6% (188)	57.3% (161)	38.6% (27)	
**Living condition**				
(Sub)urban	62.7% (220)	54.1% (152)	97.1% (68)	<0.001 ***
Rural	37.0% (130)	45.6% (128)	2.9% (2)	
Unknown	0.3% (1)	0.4% (1)	0.0% (0)	
**Role**				
Government	86.3% (303)	97.2% (273)	42.9% (30)	<0.001 ***
Non-government	12.8% (45)	2.1% (6)	55.7% (39)	
Unknown	0.9% (3)	0.7% (2)	1.4% (1)	
**Role (detailed)**				
Municipal and regional	52.1% (183)	64.1% (180)	4.3% (3)	<0.001 ***
Province	18.2% (64)	22.4% (63)	1.4% (1)	
National	7.1% (25)	4.6% (13)	17.1% (12)	
Non-government and unknown	22.5% (79)	8.9% (25)	77.1% (54)	

^A^ Country abbreviations, as follows: AUS (Australia), ESP (Spain), IND (India), MMR (Myanmar), NGA (Nigeria), and NLD (The Netherlands).

**Table 2 antibiotics-10-01486-t002:** Cumulative median and mean scores per assessment on AMR knowledge, attitude, and/or perception for all participants, stratified for high-income country (HIC) and low- and middle-income (LMIC) country participants. All scores had a maximum score of 10.

	All Participants	HICs	LMICs	Significance ^a^
**Personal knowledge**				
Means ± standard deviation	5.95 ± 2.82	5.81 ± 2.79	6.43 ± 3.57	0.053
Median [inter quartile range]	6.43 [3.57]	6.55 [2.87]	7.14 [3.57]	0.044 *
**Personal attitude and perception (AP)**				
Means ± standard deviation	6.99 ± 2.55	7.31 ± 2.38	5.70 ± 2.80	<0.001 ***
Median [inter quartile range]	7.50 [2.50]	7.50 [2.50]	5.83 [4.58]	<0.001 ***
**Political KAP**				
Means ± standard deviation	2.88 ± 2.21	2.84 ± 2.16	3.04 ± 2.14	0.529
Median [inter quartile range]	2.31 [3.08]	2.31 [3.08]	2.69 [3.37]	0.676

^a^ Significance is determined by the two-sample *t*-test for the comparison between the means and by the Mann-Whitney U-test for the comparison of the median. Significance: *** *p* < 0.001, and * *p* < 0.05.

**Table 3 antibiotics-10-01486-t003:** Multivariate logistic regression (adjusted odds ratio, aOR) of good and fair knowledge and demographic variables.

Variable	N	Good Score ^A^ % (N)	aOR 95% CI	*p*-Value	Fair Score ^A^ % (N)	aOR 95% CI	*p*-Value
	351	41.3% (156)			74.1% (260)		
**Gender ^B^**							
Female	182	44.0% (88)	ref		72.5% (132)	ref	
Male	163	44.2% (72)	1.06 [0.66–1.69]	0.805	69.9% (114)	0.96 [0.57–1.64]	0.892
**Age group ^B^**							
<40	94	41.5% (39)	ref		71.3% (67)	ref	
40–60	179	49.2% (88)	2.12 [1.11–4.04]	**0.023**	77.1% (138)	1.76 [0.85–3.64]	0.125
>60	78	37.2% (29)	2.03 [0.91–4.52]	0.080	60.3% (47)	1.68 [0.70–4.01]	0.243
**Country class ^B^**							
HIC	281	42.4% (119)	ref		70.5% (198)	ref	
LMIC	70	52.9% (37)	1.97 [0.84–4.62]	0.117	77.1% (54)	1.47 [0.52–4.10]	0.467
**Nationality ^C^**							
The Netherlands	171	43.3% (73)	ref		64.9% (111)	ref	
Spain	97	39.2% (38)	0.44 [0.23–0.86]	**0.017**	78.4% (76)	1.07 [0.51–2.25]	0.856
Myanmar	34	26.5% (9)	0.65 [0.22–1.92]	0.432	64.7% (22)	1.39 [0.42–4.64]	0.593
**Duration ^B^**							
<3 years	133	45.1% (60)			71.4% (95)	ref	
3–10 years	129	44.2% (57)	0.86 [0.50–1.48]	0.593	72.1% (93)	0.90 [0.48–1.67]	0.728
>10 years	89	43.8% (39)	0.66 [0.35–1.21]	0.178	71.9% (64)	0.65 [0.32–1.33]	0.239
**Education ^B^**							
Master and PhD	157	54.8% (86)	ref		79.6% (125)	ref	
Bachelor	143	41.3% (59)	0.61 [0.37–0.99]	0.045	75.5% (108)	0.86 [0.48–1.53]	0.597
Lower levels	49	22.5% (11)	0.25 [0.11–0.57]	**<0.001**	38.8% (19)	0.16 [0.07–0.37]	**<0.001**
**Expertise ^B^**							
Scientific	163	55.2% (90)	ref		83.4% (136)	ref	
Other	188	35.1% (66)	0.49 [0.31–0.79]	**0.004**	61.7% (116)	0.34 [0.19–0.56]	**<0.001**
**Living condition ^B^**							
(Sub)urban	220	48.2% (106)	ref		75.9% (167)	ref	
Rural	130	37.7% (49)	0.79 [0.47–1.31]	0.357	64.6% (84)	0.73 [0.41–1.30]	0.287
**Occupation ^B^**							
Government	303	37.3% (113)			71.6% (217)	ref	
Non-government	45	44.4% (20)	0.56 [0.23–1.34]	0.190	71.1% (32)	0.52 [0.18–1.46]	0.213
**Role (detailed) ^D^**							
Regional	183	38.3% (70)			66.1% (121)	ref	
Province	64	50.0% (32)	1.27 [0.69–2.36]	0.441	76.6% (49)	1.28 [0.62–2.68]	0.504
National	25	56.0% (14)	1.50 [0.57–3.95]	0.409	80.0% (20)	1.28 [0.40–4.12]	0.680
Non-government	79	50.6% (40)	1.10 [0.51–2.39]	0.804	78.5% (62)	1.27 [0.47–3.44]	0.644

^A^ Missing and unknown data are not shown in the table, and, therefore, total count does not always equal 351. ^B^ Multivariate analysis based on gender, age group, time at current role (duration), country class (HIC or LMIC), living condition, education, field of expertise, and occupation (government or non-government). ^C^ Only participants from The Netherlands, Spain, and Myanmar were included. Multivariate analysis, similar to B, excluding country class (HIC or LMIC). ^D^ Similar to B, excluding occupation (government or non-government). The numbers in bold indicate stastistical significance.

**Table 4 antibiotics-10-01486-t004:** Multivariate logistic regression (adjusted odds ratio, aOR) of good as well as fair personal attitudes and perception (AP) and demographic variables.

Variable	N	Good AP ^A^ % (N)	aOR 95% CI	*p*-Value	Fair AP ^A^ % (N)	aOR 95% CI	*p*-Value
	351	41.6% (146)			83.8% (294)		
**Gender ^B^**							
Female	182	62.1% (113)	ref		81.3% (148)	ref	
Male	163	55.2% (90)	0.76 [0.47–1.21]	0.246	85.9% (140)	1.57 [0.80–3.09]	0.194
**Age group ^B^**							
<40	94	45.7% (43)	ref		71.3% (67)	ref	
40–60	179	64.8% (116)	1.48 [0.78–2.78]	0.227	90.5% (162)	2.38 [0.96–5.90]	0.062
>60	78	59.0% (46)	1.85 [0.84–4.11]	0.129	83.3% (65)	2.46 [0.78–7.76]	0.125
**Country class ^B^**							
HIC	281	63.0% (177)	ref		88.3% (248)	ref	
LMIC	70	40.0% (28)	0.33 [0.14–0.75]	**0.009**	65.7% (46)	0.19 [0.06–0.60]	**0.005**
**Nationality ^C^**							
The Netherlands	171	56.7% (97)	ref		85.4% (146)	ref	
Spain	97	72.2% (70)	1.77 [0.90–3.45]	0.100	92.8% (90)	1.87 [0.63–5.58]	0.259
Myanmar	34	14.7% (5)	0.15 [0.05–0.52]	**0.003**	38.2% (13)	0.15 [0.04–0.57]	**0.005**
**Duration ^B^**							
<3 years	133	60.2% (80)	ref		82.7% (110)	ref	
3–10 years	129	55.8% (72)	1.06 [0.61–1.84]	0.832	83.7% (108)	1.82 [0.81–4.09]	0.148
>10 years	89	59.6% (53)	0.71 [0.38–1.33]	0.288	85.4% (76)	0.89 [0.36–2.19]	0.800
**Education ^B^**							
Master/PhD	157	61.2% (96)	ref		88.5% (139)	ref	
Bachelor	143	59.4% (85)	0.85 [0.51–1.43]	0.547	82.5% (118)	0.45 [0.21–0.97]	**0.043**
Lower levels	49	49.0% (24)	0.47 [0.22–0.99]	**0.048**	75.5% (37)	0.18 [0.06–0.53]	**0.002**
**Expertise ^B^**							
Scientific	163	67.5% (110)	ref		91.4% (149)	ref	
Other	188	50.5% (95)	0.37 [0.23–0.62]	**<0.001**	77.1% (145)	0.23 [0.10–0.50]	**<0.001**
**Living condition ^B^**							
(Sub)urban	220	59.1% (130)	ref		82.7% (182)	ref	
Rural	130	57.7% (75)	0.72 [0.43–1.21]	0.211	85.4% (111)	0.74 [0.33–1.63]	0.451
**Occupation ^B^**							
Government	303	61.4% (186)	ref		86.8% (263)	ref	
Non-government	45	18.0% (40)	0.73 [0.30–1.75]	0.483	62.2% (28)	0.43 [0.14–1.27]	0.127
**Role (detailed) ^D^**							
Regional	183	61.8% (113)	ref		88.5% (162)	ref	
Province	64	60.9% (39)	0.79 [0.43–1.48]	0.453	89.1% (57)	0.54 [0.19–1.50]	0.237
National	25	60.0% (15)	1.21 [0.44–3.30]	0.715	88.0% (22)	0.70 [0.15–3.22]	0.644
Non-government	79	48.1% (38)	0.88 [0.38–2.03]	0.765	67.1% (53)	0.18 [0.05–0.61]	**0.006**

^A^ Missing and unknown data are not shown in the table, and, therefore, total count does not always equal 351. ^B^ Multivariate analysis based on gender, age group, time at current role (duration), country class (HIC or LMIC), living condition, education, field of expertise, and occupation (government or non-government). ^C^ Only participants from The Netherlands, Spain, and Myanmar were included. Multivariate analysis similar to B, excluding country class (HIC or LMIC). ^D^ Similar to B, excluding occupation (government or non-government). The numbers in bold indicate stastistical significance.

## Data Availability

The raw data can be requested from the corresponding author.

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
