# Peer review of "Understanding Antimicrobial Resistance from the Perspective of Public Policy: A Multinational Knowledge, Attitude, and Perception Survey to Determine Global Awareness"

_antibiotics, 2021, doi:10.3390/antibiotics10121486_

Round 1

Reviewer 1 Report

Thank you authors for your manuscript. There are a few ares that requires addressing.

  1. Extra period on line 91.
  2. Line 132, Table 2. The median score for Personal knowledge for HIC participants is listed as average and standard deviation? Or is the +/- sign suppose to be brackets [ ] to signify IQR.  A mismatch is also occurring in the same row with LMIC and means +/- standard deviation. The score is listed as Median and [IQR]. The presentation of IQR is different from Table 1. I am unable to determine if this was an error in presentation and standard deviation was used of the format was Q3-Q1. Please standardize and clarify.
  3. Line 149, Figure 2. The legend requires further explanation that [Yes] was strongly agree and agree, while [No] equates to disagree and strongly disagree.
  4. Line 135-148, in section 2.3 Knowledge assessment. Please include then number of participants that indicated neutral response and number of participants that omitted response.  
  5. Throughout the paper and especially in Table 3 (line 165), be consistent with how you want to represent the p-value of either 0.805 or 0,805. Do not mix the two formats. 
  6. Line 190, Figure 3. Expand/clarify  the legend or the graph to indicate what the bar graph means, either agree/strongly agree or disagree/strongly disagree.
  7. In the limitations section, please include that the limited number participants who completed the survey will limit generalizability of this data. An example of this limitation would be the number of responses seen in figure 3.
  8. The percent and confidence intervals in figure 3 are too light to read. I suggest a darker color.
  9. Line 223, Figure 4 and Line 239, Figure 5. The legend needs to be clarified regarding what the bars represent, agree & strongly agree.

Author Response

Response:
Thank you for your positive feedback. All the comments are addressed in the revision of the
manuscript.

Thank you authors for your manuscript. There are a few ares that requires addressing.

1. Extra period on line 91.

2.
Line 132, Table 2. The median score for Personal knowledge for HIC participants is listed as average and standard deviation? Or is the +/- sign suppose to be brackets [ ] to signify IQR. A mismatch is also occurring in the same row with LMIC and means +/- standard deviation. The score is listed as Median and [IQR]. The presentation of IQR is different from Table 1. I am unable to determine if this was an error in presentation and standard deviation was used of the format was Q3-Q1. Please standardize and clarify.
3. Line 149, Figure 2. The legend requires further explanation that [Yes] was strongly agree and agree, while [No] equates to disagree and strongly disagree.

4. Line 135-148, in section 2.3 Knowledge assessment. Please include then number of participants that indicated neutral response and number of participants that omitted

5.
Throughout the paper and especially in Table 3 (line 165), be consistent with how you want to represent the p-value of either 0.805 or 0,805. Do not mix the two formats.
6. Line 190, Figure 3. Expand/clarify the legend or the graph to indicate what the bar graph means, either agree/strongly agree or disagree/strongly disagree.

7. In the limitations section, please include that the limited number participants who completed the survey will limit generalizability of this data. An example of this limitation would be the number of responses seen in figure 3.

8.
The percent and confidence intervals in figure 3 are too light to read. I suggest a darker color.
9. Line 223, Figure 4 and Line 239, Figure 5. The legend needs to be clarified regarding what the bars represent, agree & strongly agree.

response.
The main body text has been changed incorporating this. A supplementary table with a detailed overview has been added as well (please see Supplementary file 3).

Reviewer 2 Report

This paper presents an analysis of the knowledge, perception and attitude, both from a personal and political point of view, of policy makers and stakeholders from multiple countries. The authors prepared and distributed a survey and analyzed and compared the data obtained. The data is analyzed based on multiple sociodemographic factors, but the effect of the wealth of the country of origin (divided in low and medium income countries (LMIC) or high income countries (HIC)) seems to be the main target for the comparison presented in the study. The study contains very interesting and important data in the context of fighting the AMR outbreak but some details, especially on the way the data is discussed, need to be clarified and modified.

Major points:

  • The authors compare answers from HIC and LMIC groups but they have 281 responses from the first group and only 70 from the second. Shouldn’t this be more balanced in order to make the wealth of the country the main comparison factor in the study? Also, if almost 50% of participants are from the Netherlands, almost 30 % from Spain and almost 10% from Myanmar isn’t this study mainly comparing these 3 countries? This may be very important especially in the political knowledge, attitude, and perception answers that may be strongly dependent on the country policies. The authors briefly describe this bias but still generalize, by comparing results for HIC (95% Netherlands + Spain) and LMIC (77% Myanmar + India + Nigeria). This “country bias” should be clearly discussed.
  • Additionally, sociodemographic descriptors as the educational background, expertise and living condition are also significantly different between the HIC and LMIC samples. This is having an effect on the answers and consequent scores obtained (as shown in tables 3 and 4). For example, scientific expertise and higher educational background is significantly higher between the participants from LMIC and this has an effect on the larger personal knowledge scores obtained for this group. Actually, from table 3, we can see that the country class is not significantly associated with a good personal knowledge. The authors briefly refer this in the discussion and analyze these variants in table 3 and 4, but the study still focus mainly on the comparison between HIC and MLIC and, with these unbalanced samples, this should not be the main focus.
  • Is it fair to consider that a more frequent antibiotic use in the LMIC population, compared to HIC, is due to a poorer perception and attitude towards AMR? Couldn’t this only be the result of more frequent bacterial infections? This should be discussed.
  • In the discussion the authors conclude “Even after adjusting for other factors including education and field of expertise, country of origin (i.e., HIC LMIC) remained significantly associated with a good perception and behavior, suggesting that the country’s wealth status played an important role in shaping the perception and attitude towards antimicrobials.” This is true, but is it not important to say that the effect of expertise is the largest one?
  • Abstract: “HIC participants particularly emphasized the contribution of the veterinary sector to AMR, as well as the dissemination of antimicrobials in the environment”. I do not understand which results support this conclusion. I see a more positive response from LMIC when they were questioned about the use of antibiotics in animal husbandry (93% LMIC VS 79 % HIC). If this was concluded based on the answer to: “Plans to tackle AMR are mostly focusing on human health and are not addressing the livestock”, this is related with the policies of the countries but not the perspective of the participants on the problem. Again, this could be strongly biased by the fact that the vast majority of the HIC participants are politics and the majority of the LMIC participants are not.

Minor points:

“A previous study in Singapore has shown that awareness of AMR among policy makers and stakeholders can compromise the progress of AMR initiatives regardless of resource allocation” – clarify this sentence because it seems contradictory

The meaning of AMU should be added the first time this abbreviation is used

Section 2.2. The order the results are discussed is confusing. Should be the same order that is listed in table 2. The way the authors refer to the three different variants tested should also be uniform (e.g. Personal knowledge vs knowledge score).

The authors claim that spearman correlation showed an inverse correlation between personal knowledge and personal attitude and perception (table 2). If this is true, it is not perceptible from the numbers presented in table 2, the authors should provide the spearman correlation plots at least in supplementary information.

Table 2. Correct the errors of the personal knowledge mean for LMIC and median for HIC.

The parameters used to define good and fair scores are described in the methods, but they should be, at least, briefly described in the results, before table 3 is introduced and discussed.

Page 12 line 284, “On the other hand, participants from HIC had higher scores on attitude and practice, while their knowledge levels were significantly lower than participants from LMICs.” This idea is repeated before

Page 14 line 373 “in this study” is repeated.

Author Response

Response:
Thank you for your positive feedback. All the comments are addressed in the revision of the
manuscript.

This paper presents an analysis of the knowledge, perception and attitude, both from a personal
and political point of view, of policy makers and stakeholders from multiple countries. The authors
prepared and distributed a survey and analyzed and compared the data obtained. The data is
analyzed based on multiple sociodemographic factors, but the effect of the wealth of the country
of origin (divided in low and medium income countries (LMIC) or high income countries (HIC))
seems to be the main target for the comparison presented in the study. The study contains very
interesting and important data in the context of fighting the AMR outbreak but some details,
especially on the way the data is discussed, need to be clarified and modified.

Major points:

The authors compare answers from HIC and LMIC groups but they have 281 responses
from the first group and only 70 from the second. Shouldn’t this be more balanced in order
to make the wealth of the country the main comparison factor in the study? Also, if almost
50% of participants are from the Netherlands, almost 30 % from Spain and almost 10%
from Myanmar isn’t this study mainly comparing these 3 countries? This may be very
important especially in the political knowledge, attitude, and perception answers that may
be strongly dependent on the country policies. The authors briefly describe this bias but
still generalize, by comparing results for HIC (95% Netherlands + Spain) and LMIC (77%
Myanmar + India + Nigeria).
This “country bias” should be clearly discussed.
Thanks for pointing this out. We agree with this limitation of “country bias”, and we
addressed this in the methodological consideration section of the discussion (line 525 –
573). Additionally, we have added one row in Table 1 to show the top 3 countries
representation in the total sample and the HIC and LMIC cohorts.

Despite our best effort in reaching out to politicians and policymakers in different
countries, the response rate was low probably since some countries were battling with the
second wave of the Covid-19 during the survey period. The unbalanced sample size was
discussed in the manuscript, and the response rate was generally lower in LMICs which
may reflect the operational challenge in government framework in these regions. For
example, in the case of Myanmar, the parliament members and some policy makers
submitted their responses before the military coup in February, but this would not be
possible anymore due to the current political turmoil. This highlights the challenge in
conducting research studies including policymakers and politicians from LMIC.

Additionally, sociodemographic descriptors as the educational background, expertise and
living condition are also significantly different between the HIC and LMIC samples. This
is having an effect on the answers and consequent scores obtained (as shown in tables
3 and 4). For example, scientific expertise and higher educational background is
significantly higher between the participants from LMIC and this has an effect on the
larger personal knowledge scores obtained for this group. Actually, from table 3, we can
see that the country class is not significantly associated with a good personal knowledge.
The authors briefly refer this in the discussion and analyze these variants in table 3 and
4, but the study still focus mainly on the comparison between HIC and MLIC and, with
these unbalanced samples, this should not be the main focus.

Thank you for pointing this out. The effect of education and background were briefly
discussed in the result section, but we have elaborated more on this in the result and
discussion section of the revised version (please see line 458 – 465).

Is it fair to consider that a more frequent antibiotic use in the LMIC population, compared
to HIC, is due to a poorer perception and attitude towards AMR? Couldn’t this only be the
result of more frequent bacterial infections?
This should be discussed.

We agree with your comment. This perspective has now been added to our revised
manuscript (please see line 411 – 414).

In the discussion the authors conclude “Even after adjusting for other factors including
education and field of expertise, country of origin (i.e., HIC LMIC) remained significantly
associated with a good perception and behavior, suggesting that the country’s wealth
status played an important role in shaping the perception and attitude towards
antimicrobials.” This is true, but is it not important to say that the effect of expertise is the
largest one?

Thank you for pointing this out. The effect of education and background were briefly
discussed in the result section, but we have elaborated more on this in the result and
discussion section of the revised version (please see line 458 – 465).

Abstract: “HIC participants particularly emphasized the contribution of the veterinary
sector to AMR, as well as the dissemination of antimicrobials in the environment”. I do
not understand which results support this conclusion. I see a more positive response from
LMIC when they were questioned about the use of antibiotics in animal husbandry (93%
LMIC VS 79 % HIC). If this was concluded based on the answer to: “Plans to tackle AMR
are mostly focusing on human health and are not addressing the livestock”, this is related
with the policies of the countries but not the perspective of the participants on the
problem. Again, this could be strongly biased by the fact that the vast majority of the HIC
participants are politics and the majority of the LMIC participants are not.

Thank you for your feedback. We have changed the abstract accordingly based on your
comments.The existing differences in sociodemographic descriptors between HIC and
LMIC subsets have been addressed in the discussion and we have therefore chosen to
remove the mentioned sentences and replace this with our important finding that
background and education are associated with the personal and political KAP (please see
the revised abstract).

Minor points:

Thank you for your careful look at our manuscript and detailed feedback. All the comments are
addressed in the revision of the manuscript.

“A previous study in Singapore has shown that awareness of AMR among policy makers and
stakeholders can compromise the progress of AMR initiatives regardless of resource allocation”
– clarify this sentence because it seems contradictory

The meaning of AMU should be added the first time this abbreviation is used

Section 2.2. The order the results are discussed is confusing. Should be the same order that is
listed in table 2. The way the authors refer to the three different variants tested should also be
uniform (e.g. Personal knowledge vs knowledge score).

The authors claim that spearman correlation showed an inverse correlation between personal
knowledge and personal attitude and perception (table 2). If this is true, it is not perceptible from
the numbers presented in table 2, the authors should provide the spearman correlation plots at
least in supplementary information.

The coefficients of the spearman correlation were indeed correct, but we considered this to be of
no/low added value for the overall manuscript. Thanks for pointing this out and we have decided
to remove this from the text and table.

Table 2. Correct the errors of the personal knowledge mean for LMIC and median for HIC.

The parameters used to define good and fair scores are described in the methods, but they should
be, at least, briefly described in the results, before table 3 is introduced and discussed.

Page 12 line 284, “On the other hand, participants from HIC had higher scores on attitude and
practice, while their knowledge levels were significantly lower than participants from LMICs.” This
idea is repeated before

We moved this sentence to the one after the sentence discussing “While LMIC participants
displayed poor attitude and practice, they demonstrated to have a better knowledge on AMR than
HIC participants.” (line 406 – 407)

Page 14 line 373 “in this study” is repeated.

Reviewer 3 Report

Concerns:

  • The article is of low impact with reference to journal “antibiotics”. Data is less than sufficient to fulfil the requirements of journal. Although the study is funded but seems that more data is required to draw some valid conclusions in favor of spread of antimicrobial resistance

Authors are suggested to relook into article by corelating with spread of AMR. In this context more data should be added that can serve the purpose of research article.

  • How did you justify greater variations of sampling between high income (n=281) and low-income countries (n=70)? While the low-income countries are more in the world than high income countries. Moreover, there is more political instability in low-income countries that can directly affect what is discussed in this article. High income countries are aware of what is happening, and their population are influential on their government.

Moreover, except Netherlands and Spain, none of the participants could cross double figure. Statistical inferences become very questionable on such data

  • Data is expended in larger tables while the actual information can only be narrated in one table. e.g table 1 and table 2 does not drawing any important information that can be related with AMR

Round 2

Reviewer 3 Report

Sample size should not be low, while keeping in view the constraints of COVID-19 as significant barrier, I accept your suggestion.